# Predicting the Impact of Climate Change on the Geographical Distribution of Leafhopper, *Cicadella viridis* in China through the MaxEnt Model

**DOI:** 10.3390/insects14070586

**Published:** 2023-06-28

**Authors:** Xinju Wei, Danping Xu, Zhihang Zhuo

**Affiliations:** College of Life Science, China West Normal University, Nanchong 637002, China; weixinjuxx@foxmail.com (X.W.); xudanping@cwnu.edu.cn (D.X.)

**Keywords:** *Cicadella viridis*, MaxEnt, climate change, potentially suitable distribution, environment variables

## Abstract

**Simple Summary:**

*Cicadella viridis* is a polyphagous pest that feeds on plant sap. During its feeding or oviposition periods, *C. viridis* can cause significant economic losses to fruit trees, willows, and rice crops. This study utilized the maximum entropy (MaxEnt) model in species distribution modeling to predict the potential suitable distribution of *C. viridis* in China. The results indicate that climate change has had a significant impact on the geographic distribution of the leafhopper, particularly by reducing the high and moderate suitability zones (except for the RCP8.5 scenario in the 2090s). Furthermore, an analysis of several key environmental variables revealed that temperature and precipitation may be important influencing factors in the selection of suitable habitats for the leafhopper. This study provides important guidance for the future development of effective monitoring and control methods for *C. viridis*.

**Abstract:**

*Cicadella viridis* (Hemiptera: Cicadellidae) is an omnivorous leafhopper that feeds on plant sap. It significantly reduces the yield of agricultural and forestry crops while feeding or ovipositing on the host plant. In recent years, the rapid expansion of *C. viridis* has posed a serious threat to agricultural and forestry crops. To study the impact of climate change on the geographical distribution of the leafhopper, the maximum entropy (MaxEnt) model and ArcGIS software, combined with 253 geographic distribution records of the pest and 24 environmental variables, were used, for the first time, to predict the potential distribution of *C. viridis* in China under conditions of climatic change. The results showed that the currently suitable areas for *C. viridis* are 29.06–43° N, 65.25–85.15° E, and 93.45–128.85° E, with an estimated area of 11,231,423.79 km^2^, i.e., 11.66% of China. The Loess Plateau, the North China Plain, and the Shandong Peninsula are the main suitable areas. The potential distribution of the leafhopper for the high and medium suitability areas decreased under each climate scenario (except RCP8.5 in the 2090s). Several key variables that have the most significant effect on the distribution of *C. viridis* were identified, including the mean annual temperature (Bio1), the standard deviation of temperature seasonality (Bio4), the minimum temperature of the coldest month (Bio6), and the precipitation of the coldest quarter (Bio19). Our research provides important guidance for developing effective monitoring and pest control methods for *C. viridis*, given the predicted challenges of altered pest dynamics related to future climate change.

## 1. Introduction

The biggest biological population on Earth is made up of insects, with herbivorous species comprising about 50% of all insects and causing 18% of global agricultural damage [1]. This population can cause significant economic losses for many crops, including fruit trees, willows, and rice crops. The damage of *Cicadella viridis* to trees is mainly related to two aspects: on one hand, adults and nymphs feed on xylem and phloem juice [2], harming fruit trees and seedlings, thus affecting the normal growth and development of trees, leading to dry branches or the death of the whole plant [3,4]; and on the other hand, after the adults and nymphs have laid their eggs or fed on the trees, they form regular crescent-shaped trails, which represent tissue necrosis. If the number of leafhoppers is too large, they will aggravate the loss of water from the host branches, causing the plant to die. Insects are well-known vectors of viruses, and *C. viridis* is one of the known plant virus carriers [5]. More than 100 pathogens that can cause serious plant diseases can be transmitted through *C. viridis* [6]. Given that the wound made by *C. viridis* is located beneath the trunk’s cortex and the branch’s epidermis, it could potentially transmit viruses during growth and lead to further disease in the branch. In serious cases, it may cause the seedlings or delicate branches to wither and die, and it may also cause the trees to not grow regularly until they die [5,7]. In the past few years, *C. viridis* has become a major pest problem due to its collective harm, and its pest and disease issues have become increasingly prominent as its range area expands [7,8]. Thus, it is urgently necessary for scientific research into *C. viridis* to assist in establishing a clear and definite distribution of its probable adaptation zones and contribute to the creation of efficient control strategies.

The Cicadellidae is an important insect family containing many species that often cause significant damage to plant leaves. One species of this family, i.e., *C. viridis*, is of particular importance, it being an omnivorous pest that feeds on plant sap. It is widely distributed globally, as its range includes the eastern part of the Palaearctic Ecoregion, the Nearctic ecozone, the Near East, most of Europe, and the Oriental realm [3]. Many researchers have demonstrated that it is possible to infer a species’ potential geographic distribution by examining the regional climate, which is very helpful for ecological assessment [9]. Modern computer-aided technologies, in particular, have become key technologies for predicting the potential geographical distribution of species [10]. A technique for simulating the geographic distribution and ecological needs of species is known as a species distribution model (SDMs), which is based on data on species’ distribution and pertinent environmental parameters [11]. SDMs and ecological needs based on known species distribution data and associated environmental factors [11,12] have been extensively used in conservation biology, biogeography, ecology, evolutionary biology, and other fields [13]. They are a key tool for biodiversity and conservation. Among the many SDMs available, MaxEnt is a piece of software used to simulate species distribution from a record of species based on the Maximum Entropy Model [12,14]. The MaxEnt model creates background points at random using a maximum entropy distribution search while estimating the probability of species occurrence by examining species occurrence data [15]. To determine the likelihood of occurrence between 0 (unlikely) and 1 (most likely to occur), a distribution model is created using the average logistic output after going through 10 trials [15]. Plenty of research investigations have shown that the model performs quite well in forecasting the possible geographic distribution of species, especially when there are no data on the distribution of species; is more accurate than other models; and is, therefore, considered to be one of the most accurate methods used to predict a species’ potential geographical distribution [10]. There have been numerous studies indicating that this model demonstrates superior performance in terms of predictive accuracy, especially when species distribution data are lacking. It is one of the most accurate approaches for estimating a species’ probable geographic range, and it is more accurate than existing alternative models [16]. As a result of their quick running times, ease of operation, low sample demand, and stable outcomes [17,18], models of species distribution have been extensively used to forecast the spread of invasive species [19,20], the potential geographic distribution of endangered plants [13,21], changes in wild animal habitats [22], etc.; these model have been used to study species such as *Solenopsis invicta* [9] and *Spodoptera frugiperda* [23], *Isoetes* [13] and *Cunninghamia lanceolata* [21], giant panda [22], etc. Since the discovery of *C. viridis*, many scholars at home and abroad have studied its gene sequencing [24], virus transmission [2], molecular structure [25], biological characteristics [6,26], natural enemy resources, and control measures [2], achieving fruitful results. However, there is relatively limited predictive research on *C. viridis*, making it necessary to predict its potential distribution to improve control, prevent its potential harm, and avoid exacerbating its damage to fruit trees, seedlings, and rice fields.

To forecast and evaluate the prospective distribution of this pest, the MaxEnt model was incorporated into geographic distribution records and high-resolution environmental data regarding *C. viridis*. For the creation of comprehensive control plans, determining the pest’s geographic spread under favorable climate conditions is essential [27]. Currently, there are few research models that predict the distribution of *C. viridis*. The MaxEnt model’s simulation of the present and prospective distribution of *C. viridis* under three different climate scenarios offers a crucial direction for further investigation of pest management strategies.

## 2. Materials and Methods

### 2.1. Species Presence Records

To simulate a species’ acceptable range using an ecological niche model, data on the target species’ current distribution must first be collected [28]. In this research, the geographic coordinates of *C. viridis* were obtained using the following techniques: (1) searching the Global Biodiversity Information Facility (GBIF, https://www.gbif.org/, accessed on 13 January 2023) and the National Specimen Information Infrastructure (NSII, http://www.nsii.org.cn/, accessed on 13 January 2023) for information on biodiversity worldwide yielded a total of 157 distribution points; and (2) a substantial amount of *C. viridis*-related literature was reviewed, yielding a total of 96 distribution points. In this paper, Google Maps was used to identify the latitude and longitude figures of distribution points. The data were then filtered following MaxEnt’s requirements to exclude redundant records, hazy records, and records with no geographic location information [29,30]. A total of 253 distribution points were collected in this work. All collected records were saved in “CSV” format.

### 2.2. Environment Variables

To ensure that the MaxEnt 3.4.4 (https://biodiversityinformatics.amnh.org/open_source/maxent/, RRID: SCR_021830, accessed on 13 January 2023) model accurately simulated the potential geographical distribution of *C. viridis*, this article selected 19 bioclimate variables, which were downloaded from the Global Climate Database (https://www.worldclim.org/, accessed on 13 January 2023). We used the IPCC concentration pathways (RCPs) as future climate change scenarios, namely RCP2.6, RCP4.5, and RCP8.5 [31,32]. RCP2.6 and RCP8.5 represented the minimum and maximum greenhouse gas emission scenarios, respectively, and RCP4.5 represented the medium greenhouse gas emission scenario, as it was superior to other medium greenhouse gas emission scenarios (RCP6.0) [33,34]. The selection of environmental variables was mainly based on the existing records of species and the spatial correlations between variables [35,36]. Not all environmental variables were necessary to assess the potential geographical range of species because there were numerous significant correlations among some of the collected environmental variables [36]. The process of building models prioritized factors that had a greater effect on the distribution of species over those that had a smaller impact [37]. The 241 distribution locations of *C. viridis* and all 19 bioclimatic variables were added to MaxEnt, and variables with low percentage contribution rates were eliminated. Next, Pearson’s correlation coefficient analysis was conducted using SPSS software to calculate the correlation between environmental factors with high contribution rates [38]. To avoid overfitting the outcome due to the strong correlation between environmental factors, the predictive model selected variables with lower relevance, and higher contribution rates [39]. Next, we reconstructed the distribution model using these critical environmental variables.

### 2.3. Modeling Process and Statistical Analysis

The relevant data regarding distribution points of *C. viridis* and environmental variables in Table 1 were imported into MaxEnt to establish the initial model. We randomly selected 75% of the data as training data and used the remaining 25% of distribution points as testing data, with this operation being repeated 10 times to establish the predictive model. Response curves were created to determine the logical relationship between the probability of distribution and climate factors after the relevance of important environmental variables was assessed using a jackknife test [40].

Using the receiver operating characteristic (ROC) curve and default values for the remaining parameters, the model’s prediction accuracy was evaluated. The MaxEnt forecast performance indicator was based on the area under the ROC (AUC) curve [41]. The average AUC values ranged from 0.5 to 1.0, and the predictive accuracy of the model increased as the AUC value approached 1 [36]. The AUC values were separated into four performance categories [42]: between 0.5 and 0.6, failing; between 0.6 and 0.7, poor; between 0.8 and 0.9, excellent; and above 0.9, superior [29,43].

Utilizing ArcGIS, the data generated via MaxEnt for China were extracted, and the climate suitability of *C. viridis* in China was examined. The distribution levels were divided based on existence probability being regarded as the unsuitable area with *p* < 0.05, the low suitability area with 0.05 ≤ *p* < 0.33, the medium suitability area with 0.33 ≤ *p* < 0.66, and the high suitability area with ≥0.66 [36].

## 3. Results

### 3.1. Model Performance and Variable Selection

According to Figure 1, the *C. viridis* regional distribution model’s AUC value is 0.807 when taking into account all environmental variables and dominant environmental variables. This result shows that the model has a good level of predicted accuracy and offers pertinent data for examining the ways in which climate change may affect the spread of *C. viridis*.

The average contribution rate of each parameter was determined using cross-validation tests to identify the significant environmental variables. The experiment results showed that the annual mean temperature (bio1), temperature seasonality (standard deviation) (bio4), minimum temperature in the coldest month (bio6), and precipitation in the coldest quarter (bio19) were the main environmental variables (Figure 2). Among these four environmental variables, bio6 (30.9%), bio19 (28.7%), and bio4 (21.9%) had a total contribution rate of 100% (Table 1), which was a great deal higher than those of the other variables. Finally, through the above steps, four climate–environmental factors (Table 2) were determined to establish the final distribution model of *C. viridis*.

### 3.2. Potential Distribution of C. viridis in the Current Period

The MaxEnt model was used to establish the current distribution prediction map for the most suitable habitat for *C. viridis*. The map has four levels of standards: highly suitable, moderately suitable, poorly suitable, and unsuitable, as shown in Figure 3. Based on simulation results of four key environmental variables and records of the occurrence of *C. viridis*, areas that are highly suitable are mainly concentrated in the Loess Plateau; the northern part of the North China Plain extending to the Bohai Sea; the central part of the Junggar Basin, near the Tian Shan Mountains; and the southwestern part of the Tarim Basin under modern climate conditions, with the highly suitable area being at the center of these areas and extending to low-suitable areas.

Table 3 shows the statistical data of the main suitable distribution areas for *C. viridis*. The outcome reveals that areas that are highly suitable are mainly concentrated in Xinjiang, Gansu, Hebei, Shanxi, Shaanxi, Shandong, Henan, Ningxia, Qinghai, Tianjin, Beijing, and Liaoning, which account for 1.18%, 1.51%, 1.29%, 1.20%, 1.55%, 1.08%, 0.47%, 0.14%, 0.13%, 0.16%, and 0.24% of the overall area of highly suitable zones in China, respectively. The total area of highly suitable zones is 112.31 × 104 km^2^, which accounts for 11.66% of the total area of China. Shanxi has the largest highly suitable area, which covers an area of 17.28 × 104 km^2^, accounting for 17.9% of the total suitable area in China. In addition, it is noteworthy that the highly suitable zone for Beijing (96.44%) and Shandong (94.54%) are approaching the total area of their respective provinces, and all of Tianjin and Shaanxi (100%) are listed as highly suitable areas.

### 3.3. Potential Distribution of C. viridis in the Future Period

Based on the predictions of the three climate change scenarios, i.e., RCP2.6, RCP4.5, and RCP8.5, the suitable distribution range of *C. viridis* during the periods of 2041–2060 and 2071–2090 is shown in Figure 4. The Loess Plateau; the northern portion of the North China Plain to the line encircling the Bohai Sea; the central portion of the Junggar Basin, close to the Tian Shan Mountains; and the southwestern portion of the Tarim Basin are the main distribution areas of this insect’s high suitability zones. Moreover, there are a few distributions in the western, southern, and eastern coastal regions of Hainan Island, as well as in the southeasterly portion of the Qaidam Basin. The estimated distribution areas for the 2050s and 2080s, however, deviate significantly from the current areas. In the 2050s, under the RCP8.5 scenario, the appropriate area will marginally grow in comparison to the present area (Table 4).

The forecasts state that the extent of high-suitability regions of *C. viridis* will drop by 5.58% and 9.02%, respectively, under the RCP2.6 and RCP4.5 scenarios by the 2050s, while in only two scenarios will the high-suitability area increase by 19.28% by the 2050s.

The low-suitability area of *C. viridis* will expand by 16.81%, 15.75%, and 13.21%, respectively, by the 2090s under the RCP2.6, RCP4.5, and RCP8.5 scenarios. The high- and medium-suitability areas would decline in this scenario.

### 3.4. Environmental Variables Affecting the Geographical Distribution of C. viridis

By examining the scenarios of “only variables”, “no variables” and “all variables,” this study employed MaxEnt software to conduct the jackknife test of environmental factors to discover the degree to which environmental variables influence the distribution of *C. viridis* (Figure 2). The findings show that bio6 and bio1 exceeded the 0.4 and 0.25 criteria, respectively, demonstrating that they are environmental factors that significantly influence the dispersion of *C. viridis*. These findings support the notion that low temperatures play a significant role in *C. viridis* distribution.

Figure 5a–c of the MaxEnt model shows the suitable ranges of bio1, bio4, and bio6 (i.e., the probability of *C. viridis* presence is ≥0.66), which are 7.18–15.10 °C (with the highest point at 10.13 °C), 869.89–1150.67 °C (with the highest point at 954.27 °C), and −14.48 °C to −2.20 °C (with the highest point at −9.62 °C), respectively (see Table 5 for details). It should be noted that, unlike other variables, the trend of the bio6 variable shows a second increase (Figure 5c), and its adaptive range (the probability of *C. viridis* presence is ≥0.66) is greater than or equal to 15.03 °C. In addition, we found that the most suitable range of bio19 is 15.82–38.25 mm (with the highest point at 19.45 mm). The primary environmental variables have a certain influence on the likelihood of *C. viridis’* existence within these scenarios of change.

## 4. Discussion

Research has demonstrated that the environmental variables used can affect the accuracy of the predicted ecological niche model [44]. The previously used MaxEnt model predicted the study of *C. viridis’* geographical distribution using 67 variables from the Bioclim data set (http://www.worldclim.org, accessed on 13 January 2023), which were calculated based on temperature and precipitation [40]. Thus, these variables have inevitable self-relevant, multi-linear repetition, as well as redundancy problems [40,45]. Highly linked variables are confirmed to add redundant information to model predictions, affecting the model’s ability to predict outcomes. Therefore, to avoid the problem of self-relevant multi-linear repetition between variables in the modeling process, the environmental variables should first be filtered [46]. The folding jackknife test and Pearson’s correlation coefficient analysis were employed in this study to figure out the relative contributions of every variable to the data on species distribution, with the less correlated, higher contributing variables being chosen [39]. In total, four main environmental variables were obtained for model reconstruction, which improved the precision of the predicted outcomes. The most popular technique for evaluating model precision was the subject operating characteristic curve (ROC), i.e., the AUC method [41]. The diagnostic threshold did not affect the AUC values, and the performance assessment could be obtained within the threshold range. As a result, it was possible to research the association between the spread of *C. viridis* and climate change using the model’s predictions.

The predictive results of the MaxEnt model indicate that *C. viridis* is currently spread in the range of 29.06–43° N, 65.25–85.15° E, and 93.45–128.85° E, with the most suitable areas being distributed in the Yellow Plateau, the vast majority of the North China Plain, and the Shandong Peninsula. Furthermore, climate change has significantly affected the geographical distribution of the leafhopper, particularly by reducing the distribution range of high- and medium-suitable areas (except for in the 2090s RCP8.5). The previous literature confirms that the extent of the distribution of *C. viridis* is associated with the extent of the plant distribution area affected, while that the scope of activity can be influenced by various factors, though this scope is limited to the habitat [47,48]. There are notable differences between various periods and climate change situations, which might be connected to variations in temperature and rainfall in various emission scenarios [16]. According to Figure 4, over time, the trend of highly suitable areas will gradually rise; the suitable area will expand to the south and continue to increase in areas such as Hubei, Anhui, Jiangsu, and Hainan. Moreover, global warming will continue to expand the livable area of *C. viridis*: it is currently predicted that the areas not suitable for the growth of blue leaf will gradually become more suitable areas. Apart from the influence of climatic factors, it can be seen from Figure 6 that the actual distribution range of the host plants of *C. viridis* is greater than that of *C. viridis*. This observation may be the reason for the future expansion of the insect to the south, especially in the 2050s under RCP 8.5. In the future, its potentially highly suitable regional distribution will be wider, seriously hindering the development of agriculture and forestry and even threatening local ecological balance [7]. Therefore, adequate attention should be given to forestry-related sectors outside of the appropriate area, and programs for the monitoring and prevention of *C. viridis* should be strengthened where necessary.

In general, the climate plays a major role in determining how different species are distributed [49]. Temperature variations are significant [50], and host plants and other aspects of an insect’s life cycle, such as reproduction, growth, and survival, are directly impacted. Since most insects are susceptible to variations in temperature, particularly a rise in cold temperatures, climate warming is the main climatic factor influencing the spread of *C. viridis* [51]. There have been reports of significant increases in populations of *Nephotettix cincticeps* and *C. viridis* in the Yangtze River basin in August and September [52]. Given that the Yangtze River basin is one of China’s primary crop-growing regions, it is clear that warming temperatures have a substantial effect on where *C. viridis* is found. During the investigation [7], it was found that nymph and adult insect occurrence numbers and temperatures may have had a certain relationship when the outdoor environment’s temperature was between 28 and 32 °C; *C. viridis* is the most active nymph and adult insect, and this period also corresponded with the insect high-rise period. This study studied the correlation between the likelihood of *C. viridis* presence and the primary environmental variable, and it produced a corresponding response curve (Figure 5). Since environmental variables exhibit collinearity, the model will filter them to obtain key environmental variables. These key environmental variables are accurate, reliable, and can be further validated, and they include *Heortia vitessoides* Moore [34], *Osphya* [53], and *Quercus libani* Olivier [54]. According to the findings, *C. viridis’* spread is primarily influenced by bio1, bio4, bio6, and bio19. [55] This observation implies that precipitation, which primarily affects air, soil, and host growth, can also obstruct insects from developing and reproducing normally. The precipitation in the coldest quarter (bio19) is an essential temperature variable that influences the distribution of *C. viridis*, with its range of fitness declining rapidly until it plateaus when rainfall reaches a peak of 19.45 mm. The minimum temperature of the coldest month (bio6) showed two upward trends, proving that the temperature rebound had some influence on the distribution of *C. viridis*. The number of generations is influenced by habitat and geographical differences in the distribution of *C. viridis* [7]. In research into *Lilioceris lilii*, where the variables influenced the net photosynthesis rate, evaporation rate, and root vitality of the plant that hosted the species, the important role of variations in precipitation were also stressed [56]. These variables indirectly affected the extent of *C. viridis*. According to the experimental findings, *C. viridis* is currently mostly found in temperate monsoon climates and temperate continental climates. With global warming, the present and prospective suitable distribution areas for *C. viridis* are mainly in subtropical and tropical monsoon climates, with a gradual expansion into the south and some northern areas. *C. viridis* was predicted to occur in Hubei, Anhui, Jiangsu, Zhejiang, Chongqing, Taiwan, Guizhou, and Tibet under future climate scenario models. This trend could make the control of *C. viridis* even more difficult. Therefore, an accurate understanding of the occurrence of *C. viridis*, as well as of its life habits, can provide a basis for predicting the local distribution of the pest and determining the time and methods of prevention.

Despite the advantages of the MaxEnt model in terms of ease of operation, limited sample size, and elevated prediction accuracy, there are inevitable limitations that are similar to those associated with alternative ecological niche prediction models [57]. The current occurrence of *C. viridis* was used in this experiment as distribution data, and the 67 environmental variables, which were largely dependent on temperature and precipitation, were set to values based on the actual distribution of climatic extremes. The influence of future distribution sites may, therefore, be ignored, and numerous other aspects, such as topography, environmental factors, soil factors, distribution of natural enemies, correlation of host plants, and interspecific factors, are not taken into account [58], which can cause some error in the model’s predictions. However, in a given model, we cannot take into account all environmental factors [16]; thus, some errors are inevitable. In this study, our focus was on the influence of climate on the distribution of *C. viridis*. Therefore, the impact of the distribution of various factors, as well as other variables, on *C. viridis* should be considered in future studies to further improve the predictive reliability of the model.

## 5. Conclusions

The MaxEnt model was successfully used in this work to examine the geographical range and environmental adaptation of *C. viridis*. The model’s results show that climate change is predicted to affect the distribution of *C. viridis*, reducing the area of highly and moderately suitable regions, except for in the 2090s RCP8.5 scenario. Xinjiang, southeastern Gansu, Shanxi, Shaanxi, Hebei, Henan, Shandong, Ningxia, Tianjin, and Beijing are predicted to be potentially suitable areas for the future growth of *C. viridis*. Additionally, our study found that *C. viridis* can survive in at least 13 provinces in China, with a suitable growth range of 29.06–43° N, 65.25–85.15° E, and 93.45–128.85° E. The environmental variables (Bio1, Bio4, Bio6, and Bio19) have a significant impact on the distribution of *C. viridis*. Overall, the results of this study provide a scientific basis for predicting the distribution and control of *C. viridis*.

## Figures and Tables

**Figure 1 insects-14-00586-f001:**
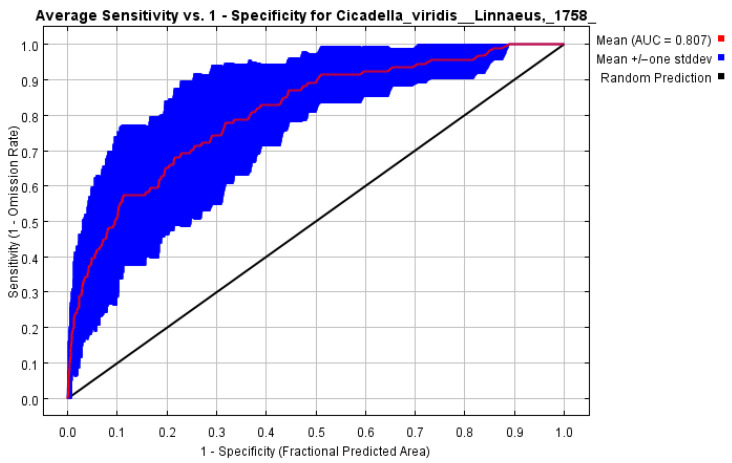
The receiver operating characteristic (ROC) curves and the area under the ROC curve (AUC) values for the studied period (1950–2000).

**Figure 2 insects-14-00586-f002:**
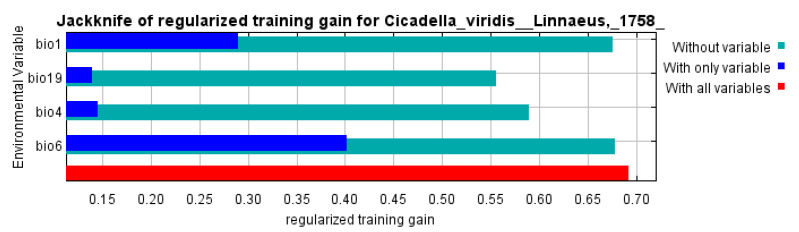
Variable importance, as determined via the folding jackknife test, for *C. viridis*.

**Figure 3 insects-14-00586-f003:**
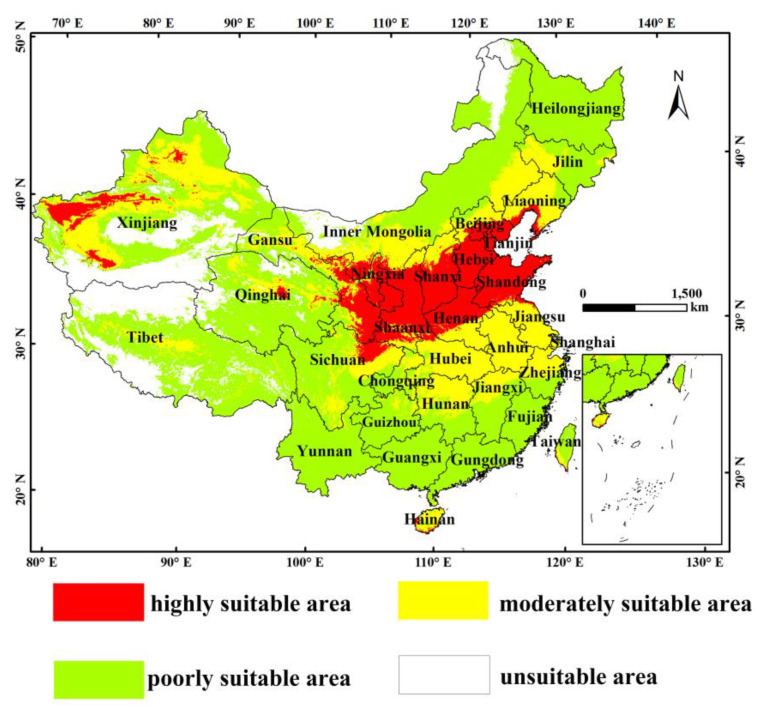
Status of the suitable climatic distribution of *C. viridis* in China.

**Figure 4 insects-14-00586-f004:**
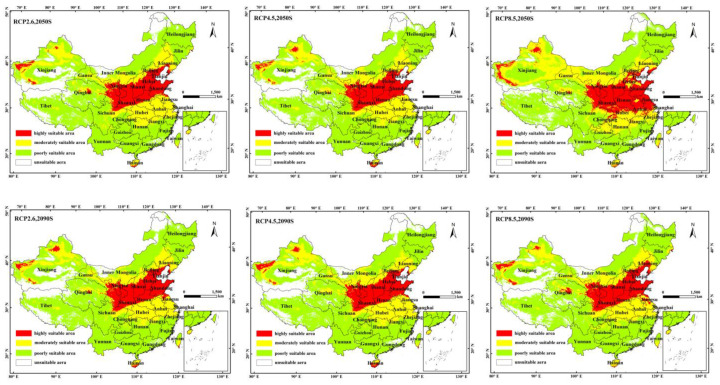
The potential distribution of *C. viridis* in suitable regions in China under different climatic conditions. The color blocks of the region show the probability of the occurrence of *C. viridis*. Red represents a probability higher than 0.66 of being a high suitability area, yellow represents a medium suitability area with a 0.33–0.66 probability, green represents low suitability areas with a 0.05–0.33 probability, and white represents unsuitable areas.

**Figure 5 insects-14-00586-f005:**
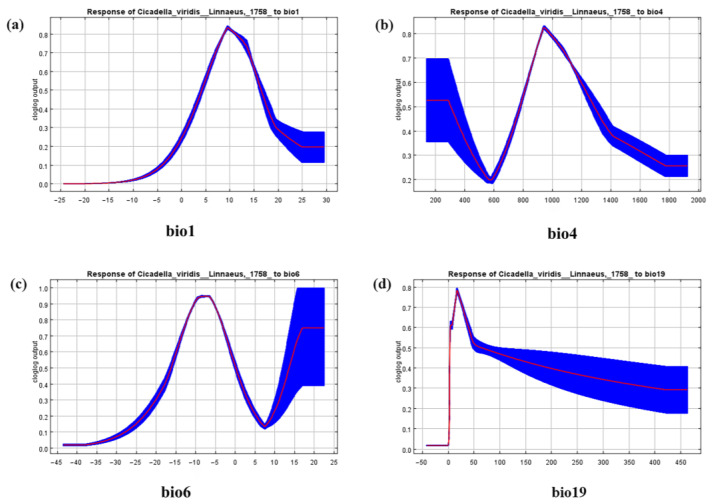
Response curves for the environmental variables that contribute most to the MaxEnt model. (**a**) The annual mean temperature (bio1). (**b**) The standard deviation of seasonal temperature variation (bio4). (**c**) The minimum temperature of the coldest month (bio6). (**d**) The precipitation of the coldest quarter (bio19). The red line is the average response of the Maxent run. The blue part is the average +/− one standard deviation.

**Figure 6 insects-14-00586-f006:**
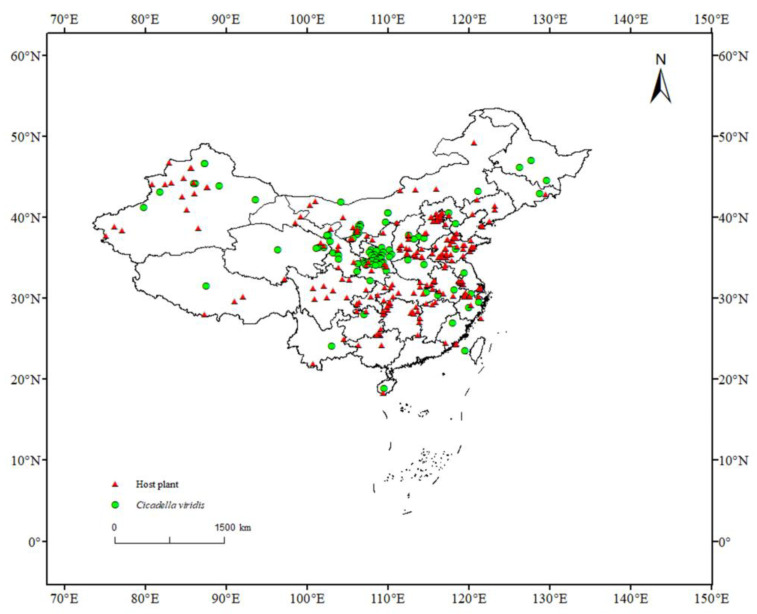
Comparison between the actual distribution of *C. viridis* and its host plants. Red triangles represent host plants, while green circles represent *C. viridis*. Data source: https://doi.org/10.6084/m9.figshare.23496806.v1, accessed on 14 January 2023.

**Table 1 insects-14-00586-t001:** Percentage contribution and ranked importance of environmental variables in the MaxEnt model.

Variable	Percent Contribution (%)	Permutation Importance (%)
Bio6	30.9	28.2
Bio19	28.7	29.6
Bio4	21.9	19.2
Bio1	18.5	23

**Table 2 insects-14-00586-t002:** Environmental variables (bioclimatic features) are retained in the modeling process.

Variable	Environmental Variables	Unit
Bio1	Mean annual temperature	°C
Bio4	Temperature seasonality	°C
Bio6	Minimum temperature of the coldest month	°C
Bio19	Precipitation of the coldest quarter	°C

**Table 3 insects-14-00586-t003:** Analysis of the main suitable distribution areas for *C. viridis*.

Province	Highly Suitable Area (10^4^ km^2^)	Total (10^4^ km^2^) *	Percentage of Highly Suitable Areas in Province (%)	Percentage of Highly Suitable Areas in China (%)
Inner Mongolia	3.73	118.30	0.0316	0.0039
Xinjiang	11.35	166.00	0.0684	0.0118
Jilin	0.00	18.74	0.0002	0.0000
Liaoning	2.27	14.80	0.1532	0.0024
Gansu	14.58	45.37	0.3213	0.0151
Hebei	12.39	18.88	0.6561	0.0129
Beijing	1.58	1.64	0.9644	0.0016
Shanxi	11.59	15.67	0.7399	0.0120
Tianjin	1.22	1.19	1.0271	0.0013
Shaanxi	17.28	15.67	1.1028	0.0179
Ningxia	4.49	6.64	0.6761	0.0047
Qinghai	1.39	72.1	0.0193	0.0014
Shandong	14.94	15.80	0.9454	0.0155
Henan	10.44	16.70	0.6250	0.0108
Jiangsu	0.69	10.72	0.0643	0.0007
Anhui	0.10	14.01	0.0069	0.0001
Sichuan	3.17	48.60	0.0652	0.0033
Hubei	0.68	18.59	0.0365	0.0007
Taiwan	0.04	3.60	0.0121	0.0000
Hainan	0.38	3.54	0.1079	0.0004
China	112.31	/	/	0.1166

* Indicates the total area of the corresponding province.

**Table 4 insects-14-00586-t004:** Predicted suitable areas for *C. viridis* under current and future climatic conditions.

Decade Scenarios		Predicted Area (km^2^)			Comparison with Current Distribution (%)		
		Poorly Suitable Aera	Moderately Suitable Aera	Highly Suitable Aera	Poorly Suitable Aera	Moderately Suitable Area	Highly Suitable Aera
Current		436.34	222.63	111.62			
2050s	RCP2.6	500.63	202.94	105.39	0.1473	−0.0884	−0.0558
	RCP4.5	496.27	193.73	101.55	0.1373	−0.1298	−0.0902
	RCP8.5	472.88	248.64	133.14	0.0837	0.1168	0.1928
2090s	RCP2.6	509.70	197.93	104.00	0.1681	−0.1109	−0.0683
	RCP4.5	505.06	198.74	102.62	0.1575	−0.1073	−0.0806
	RCP8.5	493.97	195.13	111.20	0.1321	−0.1235	−0.0038

**Table 5 insects-14-00586-t005:** Appropriate survival range and best survival point for the three variables bio1, bio4, and bio6.

Variable	Suitable Range (°C)	Best Survival Point (°C)
Bio1	7.18~15.10	10.13
Bio4	869.89~954.27	954.27
Bio6	−14.48~−2.20	−9.62

## Data Availability

The data supporting the results are available in a public repository at: GBIF.org (accessed on 13 January 2023), GBIF Occurrence Download https://doi.org/10.15468/dl.3dqd95, accessed on 13 January 2023 and Xinju Wei (2023): *Cicadella viridis* occurrence.xlsx. figshare. Dataset. https://doi.org/10.6084/m9.figshare.21897090.v1, accessed on 14 January 2023.

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
