# Peer review of "Predicting the Impact of Climate Change on the Geographical Distribution of Leafhopper, Cicadella viridis in China through the MaxEnt Model"

_insects, 2023, doi:10.3390/insects14070586_

Round 1

Reviewer 1 Report

1. Original Article

Recommendation, Minor Revision.

2. Comments to Author:

Manuscript ID: insects-2431119

Title: Predicting the Impact of Climate Change on the Geographical Distribution of Leafhopper, Cicadella viridis in China through the MaxEnt Model

Authors: Xinju We, Danping Xu, Zhihang Zhuo

3. Overview and general recommendation:

This is an interesting work on the MaxEnt model of Cicadella viridis in China under changing climate, and the manuscript is well-written to some degree. If the authors modified the following comments carefully, I think it can be accepted and published.

4. Major comments:

As a plant-based pest, the distribution of C. viridis is influenced by the host. However, the author of this article did not involve research on host distribution. I suggest that the author describe or discuss the interrelationships between the distribution of hosts and pests in the results or discussions.

5. Minor comments:

1). Page 1, line 36-37: This is an exploratory study, not theoretical, and therefore cannot provide a theoretical basis.

2). Page 3, line 111: Suggest replacing the expression "no research", which is too absolute.

3). Page 3, line 114-115: It is more of indirect evidence as to where this species could be occurring in the future and yet it would need to match with future distribution of its host, among other things. Therefore, no theoretical basis can be provided.

4). Page 3, line 127: Please describe exactly how many distribution points were finally chosen.

5). Page 5, line 204: Suggest deleting "best”.

6). Page 11, line 370: Suggest deleting “for the first time”.

7). Page 11, line 376-177: "Bio1, Bio4, Bio6, Bio19" are climatic factors within environmental factors.

Author Response

Dear Reviewer:

Thank you for your comments concerning our manuscript entitled " Predicting the Impact of Climate Change on the Geographical Distribution of Leafhopper, Cicadella viridis in China through the MaxEnt Model"(Insects- 2431119). Those comments are all valuable and very helpful for revising and improving our paper, as well as the important guiding significance to our research. We have studied the comments carefully and have made corrections which we hope to meet with approval. The main corrections in the paper and the response to the your comments are as flowing:

For Reviewer 1

Major comments,

As a plant-based pest, the distribution of C. viridis is influenced by the host. However, the author of this article did not involve research on host distribution. I suggest that the author describe or discuss the interrelationships between the distribution of hosts and pests in the results or discussions.

Author's response: Thanks for your valuable comment. A new figure has been added by us to the discussion section to describe the relationship between host and pest distribution. (Line 331-333, in the revised manuscript)

Minor comments,

1)" This is an exploratory study, not theoretical, and therefore cannot provide a theoretical basis.(line 36-37)

Author's response: Thanks for your meaningful comment. The phrase "provides theoretical research" has been corrected for "scientific guidance". (Line 38-39, in the revised manuscript)

2)" Suggest replacing the expression 'no research', which is too absolute.(line 111)

Author's response: Thanks for your meaningful comment. The sentence has been rewritten as quired. (Line 121-123, in the revised manuscript)

3). " It is more of indirect evidence as to where this species could be occurring in the future and yet it would need to match with future distribution of its host, among other things. Therefore, no theoretical basis can be provided.(line 114-115)

Author's response: Thanks for your meaningful comment. Corresponding changes have already been made to the relevant issues in the original text, and a new figure has been added by us to the discussion section to describe the relationship between host and pest distribution. (Line 331-333 in the revised manuscript)

4). " Please describe exactly how many distribution points were finally chosen.(line 127)

Author's response: Thanks for your meaningful comment. A total of 253 distribution points were finally chosen and it has been added in the revised manuscript. (Line 140, in the revised manuscript)

5). " Suggest deleting 'best'.(line 204)

Author's response: Thanks for your meaningful comment. "best" has been deleted. (Line 218, in the revised manuscript)

6). "Suggest deleting 'for the first time'.(line 370)

Author's response: Thanks for your meaningful comment. "for the first time" has been deleted. (Line 397, in the revised manuscript)

7). " 'Bio1, Bio4, Bio6, Bio19' are climatic factors within environmental factors.(line 376-377)

Author's response: Thanks for your meaningful comment. The "climate conditions and environmental factors" have been replaced by "environment variables". (Line 404-405, in the revised manuscript)

We tried our best to improve the manuscript and made some changes in the manuscript. These changes will not influence the content and framework of the paper.

We appreciate for your warm work earnestly and hope that the correction will meet with approval.

Once again, thank you very much for your comments and suggestions.

Regards,

Xinju Wei

College of Life Science, China West Normal University,

1 Shida Road, Nanchong, 637002, China

[Email] [email protected]

Reviewer 2 Report

This is an interesting study that models the current and potential changes in distribution of Leafhopper, Cicadella viridis in China with a warming climate. The study used ArcGIS and MaxEnt to model the change in the distribution by using the Jacknife test to select for important variables and analysing the model output under different climate scenarios. It concluded a predicted reduction in area of the highly and moderately suitable regions with climate warming.

The introduction is thorough in explaining the context for undertaking the modelling and leads to the approach taken for the study. Methods are given in detail though the reasons why variables were chosen and or excluded and improve model fit need to be explained. Also, the results needs work on flow as it contains methods and unnecessary sentences. Thus making it difficult for readers to interpret results or have the ability to apply these methods for other research.

There are several spelling and grammatical errors throughout this manuscript. Particularly the use of the words “besides” at line 29, “kind of” line 57, and “something like” at line 90 indicates the authors would seem to be unsure or are unclear what the results or the literature indicate.

Specific comments

Lines 19-20. Delete the following as repeating the brackets  “and is one of the representative species of Cicadellidae”

Line 21. change “pest” to “plant”.

Line 27-28 “with a total very appropriate area of” doesn’t  make sense. I think you mean “with an estimated area of”

Lines 25 and 31  are “elements” the same as “variables” please be consistent throughout the manuscript

Line 29-30 Analysing the effect of scenarios on an organism only predicts the potential distribution of the organisms in the future.

The sentence could be change to i.e. “The potential distribution of the leafhopper for the high and medium suitability areas decreased under each climate scenario (except RCP8.5 in the 2090s)”

Introduction 

Lines 43-44. The jump is too large between the first and second sentence. The second sentence forms the first sentence of the next paragraph. In the first paragraph expand on why herbivorous pests are of most concern in causing damage to cropping industries.

Then the first and sentence of the second paragraph is wordy please consider changing to

“The Cicadellidae is an important insect family containing many species that often cause significant damage to plant leaves”. One species of this family Cicadella viridis is of particular importance being an omnivorous pest that feeds on plant sap. It is widely distributed globally, which includes the eastern part of the Palaearctic Ecoregion, the Nearctic ecozone, the Near East, most of Europe, and the Oriental realm”.

Line 49 “As one of the representative species of Cicadellidae” delete as repeating above

Line 57. “Not only that, but pests are also a kind of virus disseminator”

Use of the words “kind of” is weak. Either C. viridis transmits viruses or dos not, the last part of the sentence would suggest this is true.

Also which pests, the species being studied, or all pests globally? Please alter the sentence with these considerations.

Line 65 “expands” provide references to support this statement

Line 75-76  “technique for simulating species' geographical distribution” delete as repeating.

Line 90  “distribution [9].” If its “plentiful” why only one reference, please insert more references.

Methods

Line 134 “four” though only three scenarios are presented in the results”

 Results

Section 3.1-much of this section is method, and should be moved.

Section 3.2- The following is method and not results. “The best current prediction chart for the distribution has been created using ArcGIS 204 10.8 to predict the observed events and environmental conditions based on the Maxent 205 model”

Lines 192-194-“The experiment results showed that the annual mean temperature (bio1), temperature seasonality (standard deviation) (bio4), minimum temperature of the coldest month (bio6), and precipitation of the coldest quarter (bio19) were the main environmental variables”

Please describe some validation of the results as on page 7 in this article https://doi.org/10.3390/su12072671 Also more testing needs to be conducted or presented to validate the decision to only use the four variables. Please consider running Pearson correlation analysis and variance inflation factor (VIF) as in the article  https://doi.org/10.3390/insects14050476

 “which was a great deal higher than the other variables”-how much higher 1 2 or 3 times?  There are no statistics, post-hoc tests to verify this statement, please show the results or statistics for the other variables either in the text or as a supplementary.

Figure 4-The top left heading for each figure the scale bar and the names of each region/state are hard to read. At least increase the font size of the headings as the regions/state are just readable in Figure 3.

Lines 262 “We got the environmental response curves from the MaxEnt model (Figure 5), where the trends of three variables (bio1, bio4, and bio6) are essentially identical.”  This sentence is not needed as the remaining text explains the figure and the readers can see there are curves. Just add “The MaxEnt model” before “show” at Line 264.

Conclusion

 Climate change modelling indicates observation has not happened though are predicted. Line 371 “has a significant impact on” change to “is likely” or “is predicted” “to have a significant impact on”

 The last sentence uses the word “predicted”. Consider using this more often when discussing results.

There are several spelling and grammatical errors throughout this manuscript. Particularly the use of the words “besides” at line 29, “kind of” line 57, and “something like” at line 90 indicates the authors would seem to be unsure or are unclear what the results or the literature indicate.

Other considerations to english/grammar are made in the comments and suggestions.

Author Response

Dear Reviewer:

Thank you for your comments concerning our manuscript entitled " Predicting the Impact of Climate Change on the Geographical Distribution of Leafhopper, Cicadella viridis in China through the MaxEnt Model"(Insects- 2431119). Those comments are all valuable and very helpful for revising and improving our paper, as well as the important guiding significance to our research. We have studied the comments carefully and have made corrections which we hope to meet with approval. The main corrections in the paper and the response to your comments are as flowing:

For Reviewer 2

Comments and Suggestions for Authors,

There are several spelling and grammatical errors throughout this manuscript. Particularly the use of the words “besides” at line 29, “kind of” line 57, and “something like” at line 90 indicates the authors would seem to be unsure or are unclear what the results or the literature indicate.

Author's response: Thanks for your valuable comment. Spelling and grammar have been carefully checked. As per your advice, "besides" has been deleted, and the sentence has been rewritten too (line 30-31, in the revised manuscript). The "kind of" in line 57(Line 60-61, in the revised manuscript) has been deleted, and the sentence, "Not only that, but pests are also a kind of virus disseminator, and C. viridis is a known plant virus vector", in the original text has been modified to "Insects are also a virus resource, and C. viridis is one of the known plant virus carriers". The "something like" in line 90 (line 100-101, in the revised manuscript) has been removed, and the corresponding reference literature has also been adjusted as required.

Specific comments,

1)"Delete the following as repeating the brackets 'and is one of the representative species of Cicadellidae'.(line 19-20)

Author's response: Thanks for your valuable comment. “and is one of the representative species of Cicadellidae” has been deleted. (Line 19-20, in the revised manuscript)

  • " change 'pest' to 'plant'.(line 21)

Author's response: Thanks for your valuable comment. The word "pest" has been changed to "plant". (Line 21, in the revised manuscript)

  • " 'with a total very appropriate area of' doesn’t make sense. I think you mean 'with an estimated area of'.(line 27-28)

Author's response: Thanks for your valuable advice. The "with a total very appropriate area of" has been changed to "with an estimated area of". (Line 27-28, in the revised manuscript)

  • " Lines 25 and 31 are 'elements' the same as 'variables' please be consistent throughout the manuscript.(line 25 and 31)

Author's response: Thanks for your valuable advice. The word "elements" has been changed to "variables" throughout the manuscript. (Line 25 and 31, in the revised manuscript)

  • "Analysing the effect of scenarios on an organism only predicts the potential distribution of the organisms in the future. The sentence could be change to i.e. 'The potential distribution of the leafhopper for the high and medium suitability areas decreased under each climate scenario (except RCP8.5 in the 2090s)'.(line 29-30)

Author's response: Thanks for your valuable advice. The sentence has been replaced as required. (Line 30-31, in the revised manuscript)

Introduction,

  • Line 43-44. The jump is too large between the first and second sentence. The second sentence forms the first sentence of the next paragraph. In the first paragraph expand on why herbivorous pests are of most concern in causing damage to cropping industries.

Then the first and sentence of the second paragraph is wordy please consider changing to "The Cicadellidae is an important insect family containing many species that often cause significant damage to plant leaves". One species of this family Cicadella viridis is of particular importance being an omnivorous pest that feeds on plant sap. It is widely distributed globally, which includes the eastern part of the Palaearctic Ecoregion, the Nearctic ecozone, the Near East, most of Europe, and the Oriental realm".

Author's response: Thanks for your valuable advice. The second sentence of the first paragraph has been moved to become the first sentence of the second paragraph, and the sentence has been revised according to your suggestion. References have also been adjusted. (Line 72-79, in the revised manuscript)

  • Line 49 “As one of the representative species of Cicadellidae” delete as repeating above.

Author's response: Thanks for your valuable advice. The "As one of the representative species of Cicadellidae" in the original text has been deleted. (Line 51, in the revised manuscript)

  • Line 57. “Not only that, but pests are also a kind of virus disseminator”.

Use of the words “kind of” is weak. Either C. viridis transmits viruses or dos not, the last part of the sentence would suggest this is true.

Also which pests, the species being studied, or all pests globally? Please alter the sentence with these considerations.

Author's response: Thanks for your valuable advice. In "Comments and suggestions for authors," the "kind of" in line 57 (Line 60-61, in the revised manuscript) has been deleted, and the sentence, "Not only that, but pests are also a kind of virus disseminator, and C. viridis is a known plant virus vector", in the original text has been modified to "Insects are also a virus resource, and C. viridis is one of the known plant virus carriers".

  • Line 65 “expands” provide references to support this statement.

Author's response: Thanks for your valuable advice. Examples of the dangers of the spread of C. viridis have been given. (Line 69, in the revised manuscript)

  • Line 75-76 “technique for simulating species' geographical distribution” delete as repeating.

Author's response: Thanks for your valuable advice. The "technique for simulating species' geographical distribution" has been deleted. (Line 85-86, in the revised manuscript)

  • Line 90 “distribution [9].” If its “plentiful” why only one reference, please insert more references.

Author's response: Thanks for your valuable advice. In "Comments and suggestions for authors", the "something like" in line 90 (line 100–101 in the revised manuscript) has been removed, and the corresponding reference has also been adjusted as required.

Methods,

  • Line 134 “four” though only three scenarios are presented in the results”.

Author's response: Thanks for your valuable advice. Among the four scenario models in the IPCC, RCP2.6, RCP4.5, and RCP8.5 were selected to represent low, medium, and high scenarios, respectively. These three scenario models are considered to provide better predictive outcomes for China. The original expression has also been revised. (Line 147-148, in the revised manuscript)

Results,

  • Section 3.1-much of this section is method, and should be moved.

Author's response: Thanks for your valuable advice. Some statements in Section 3.1 have been deleted. (Line 194-201, in the revised manuscript)

  • Section 3.2-The following is method and not results. “The best current prediction chart for the distribution has been created using ArcGIS 10.8 to predict the observed events and environmental conditions based on the Maxent model”.

Author's response: Thanks for your valuable advice. This sentence has been deleted. (Line 218-221, in the revised manuscript)

  • Lines 192-194- “The experiment results showed that the annual mean temperature (bio1), temperature seasonality (standard deviation) (bio4), minimum temperature of the coldest month (bio6), and precipitation of the coldest quarter (bio19) were the main environmental variables”.

Please describe some validation of the results as on page 7 in this article https://doi.org/10.3390/su12072671 Also more testing needs to be conducted or presented to validate the decision to only use the four variables. Please consider running Pearson correlation analysis and variance inflation factor (VIF) as in the article https://doi.org/10.3390/insects14050476

which was a great deal higher than the other variables”-how much higher 1 2 or 3 times?  There are no statistics, post-hoc tests to verify this statement, please show the results or statistics for the other variables either in the text or as a supplementary.

Author's response: Thanks for your valuable advice. Based on the references Xu et al. (2020), Wang et al. (2021), and Xu et al. (2019), we have determined that our research methodology is reliable. Additionally, we appreciate your suggestion, so we have included in the discussion, "Since environmental factors exhibit collinearity, the model will filter them to obtain key environmental variables. These key environmental factors are accurate, reliable, and can be further validated, such as Heortia vitessoides Moore [34], Osphya [53], and Quercus libani Olivier [54]". (Line 357-360, in the revised manuscript)

  1. Xu, D.; Li, X.; Jin, Y.; Zhuo, Z.; Yang, H.; Hu, J.; Wang, R. Influence of climatic factors on the potential distribution of pest Heortia vitessoides Moore in China. Glob. Ecol. Conserv. 2020, 23, e1107.
  2. Liu, T.; Liu, H.; Wang, Y.; Yang, Y. Climate Change Impacts on the Potential Distribution Pattern of Osphya (Coleoptera: Melandryidae), an Old but Small Beetle Group Distributed in the Northern Hemisphere. Insects. 2023, 14, 5, 476.
  3. çoban, H.O.; örücü, Ö.K.; Arslan, E.S. MaxEnt Modeling for Predicting the Current and Future Potential Geographical Distribution of Quercus libani Olivier. Sustainability. 2020, 12, 7, 2671.

  • Figure 4-The top left heading for each figure the scale bar and the names of each region/state are hard to read. At least increase the font size of the headings as the regions/state are just readable in Figure 3.

Author's response: Thanks for your valuable advice. Figure 4 has been revised.

  • Lines 262 “We got the environmental response curves from the MaxEnt model (Figure 5), where the trends of three variables (bio1, bio4, and bio6) are essentially identical.” This sentence is not needed as the remaining text explains the figure and the readers can see there are curves. Just add “The MaxEnt model” before “show” at Line 264.

Author's response: Thanks for your valuable advice. The sentence has been revised as requested. (Line 278-280, in the revised manuscript)

Conclusion,

  • Climate change modelling indicates observation has not happened though are predicted. Line 371 “has a significant impact on” change to “is likely” or “is predicted” “to have a significant impact on”.

Author's response: Thanks for your valuable advice. The phrase "has a significant impact on" has been changed to "is predicted". (Line 398, in the revised manuscript)

  • The last sentence uses the word “predicted”. Consider using this more often when discussing results.

Author's response: Thanks for your valuable advice. Using "predicted" in the discussion section. (Line 298 and 310, in the revised manuscript)

We tried our best to improve the manuscript and made some changes in the manuscript. These changes will not influence the content and framework of the paper.

We appreciate for your warm work earnestly and hope that the correction will meet with approval.

Once again, thank you very much for your comments and suggestions.

Regards,

Xinju Wei

College of Life Science, China West Normal University,

1 Shida Road, Nanchong, 637002, China

[Email] [email protected]

Round 2

Reviewer 2 Report

A much imporved mansucript with appropirate referencing and method jusitification, with imporved clarity of the figures. Just some minor comments/suggestions below.

Lines 33-36. change to 

"Our research provides important guidance for developing effective monitoring and pest control methods for C. viridis given the predicted challenges of altered pest dynmaics with future climate change ."

Line 51. Insects are also a virus resource change to “Insects are well known vectors of viruses”

The sentence at Line 161 is repeated at Line 167 and can be combined, also the categories can be explained more succinctly see suggestion below.

Line 163. Move “average AUC values, can range from 0.5 to 1.0” to after “[41]. “The” at line 161, then after “1.0” insert from Line 167 “model's performance is considered better as the AUC value approaches 1 [36].”

The paragraph could read “The average AUC values, range from 0.5 to 1.0, where the predictive accuracy of the model increases as the AUC value approaches 1 [36]. Thes AUC values are separated into five performance categories [42] of 0.5 and 0.6, failed, between 0.6 and 0.7, poor….”

Line 233 Figure 4- If the legend is shown once above the first 3 figures or below the last three figures, Then its size can be increased, and the figures would look less cluttered.

Line 373 change to “is predicted to effect the”

Is good, see above. Please sure sentneces/information is not being repeated.

Author Response

Dear Reviewer:

Thank you for your comments concerning our manuscript entitled " Predicting the Impact of Climate Change on the Geographical Distribution of Leafhopper, Cicadella viridis in China through the MaxEnt Model"(Insects- 2431119). Those comments are all valuable and very helpful for revising and improving our paper, as well as the important guiding significance to our research. We have studied the comments carefully and have made corrections which we hope to meet with approval. The main corrections in the paper and the response to your comments are as flowing:

For Reviewer 2

Comments and Suggestions for Authors,

1) Lines 33-36. change to "Our research provides important guidance for developing effective monitoring and pest control methods for C. viridis given the predicted challenges of altered pest dynmaics with future climate change."

Author's response: Thanks for your valuable comment. The sentence has been replaced as required. (Line 39-41, in the revised manuscript)

2) Line 51. Insects are also a virus resource change to “Insects are well known vectors of viruses”

Author's response: Thanks for your valuable comment. The phrase " Insects are also a virus resource " has been changed to " Insects are well known vectors of viruses ". (Line 62-63, in the revised manuscript)

3) The sentence at Line 161 is repeated at Line 167 and can be combined, also the categories can be explained more succinctly see suggestion below.

Line 163. Move “average AUC values, can range from 0.5 to 1.0” to after “[41]. “The” at line 161, then after “1.0” insert from Line 167 “model's performance is considered better as the AUC value approaches 1 [36].”

The paragraph could read “The average AUC values, range from 0.5 to 1.0, where the predictive accuracy of the model increases as the AUC value approaches 1 [36]. Thes AUC values are separated into five performance categories [42] of 0.5 and 0.6, failed, between 0.6 and 0.7, poor….”

Author's response: Thanks for your valuable comment. Based on your suggestion, revisions have been made to lines 163 and 167. (Line 181-185, in the revised manuscript)

4) Line 233 Figure 4- If the legend is shown once above the first 3 figures or below the last three figures, Then its size can be increased, and the figures would look less cluttered.

Author's response: Thanks for your valuable comment. We greatly appreciate your suggestion. Considering that each figure has a legend, it can provide readers with a more intuitive experience. Recently, a paper published in your esteemed journal (https://doi.org/10.3390/insects14050458) also employed a similar format for the figures.

5) Line 373 change to “is predicted to effect the”.

Author's response: Thanks for your valuable comment. Based on your suggestion, a revision has been made to line 373. (Line 403, in the revised manuscript)

We tried our best to improve the manuscript and made some changes in the manuscript. These changes will not influence the content and framework of the paper.

We appreciate your warm work earnestly and hope that the correction will meet with approval.

Once again, thank you very much for your comments and suggestions.

Regards,

Xinju Wei

College of Life Science, China West Normal University,

1 Shida Road, Nanchong, 637002, China

[Email] [email protected]